# Suppression of the *s*-Wave Order Parameter Near the Surface of the Infinite-Layer Electron-Doped Cuprate Superconductor Sr$_{0.9}$La$_{0.1}$CuO$_2$

**Rustem Khasanov** [1,*] **, Alexander Shengelaya** [2] **, Roland Brütsch** [3] **and Hugo Keller** [4]

[1] Laboratory for Muon Spin Spectroscopy, Paul Scherrer Institut, CH-5232 Villigen PSI, Switzerland
[2] Department of Physics, Tbilisi State University, Chavchavadze 3, GE-0128 Tbilisi, Georgia; alexander.shengelaya@tsu.ge
[3] Laboratory for Material Behaviour, Paul Scherrer Institut, CH-5232 Villigen PSI, Switzerland; roland.bruetsch@psi.ch
[4] Physik-Institut der Universität Zürich, Winterthurerstrasse 190, CH-8057 Zürich, Switzerland; keller@physik.uzh.ch
[*] Correspondence: rustem.khasanov@psi.ch

**Abstract:** The temperature dependencies of the in-plane ($\lambda_{ab}$) and out-of-plane ($\lambda_c$) components of the magnetic field penetration depth were investigated near the surface and in the bulk of the electron-doped superconductor Sr$_{0.9}$La$_{0.1}$CuO$_2$ by means of magnetization measurements. The measured $\lambda_{ab}(T)$ and $\lambda_c(T)$ were analyzed in terms of a two-gap model with mixed $s + d$-wave symmetry of the order parameter. $\lambda_{ab}(T)$ is well described by an almost pure anisotropic $d$-wave symmetry component ($\simeq 96\%$), mainly reflecting the surface properties of the sample. In contrast, $\lambda_c(T)$ exhibits a mixed $s + d$-wave order parameter with a substantial $s$-wave component of more than 50%. The comparison of $\lambda_{ab}^{-2}(T)$ measured near the surface with that determined in the bulk by means of the muon-spin rotation/relaxation technique demonstrates that the suppression of the $s$-wave component of the order parameter near the surface is associated with a reduction of the superfluid density by more than a factor of two.

**Keywords:** superconductivity; cuprates; magnetic penetration depth; order parameter; superconducting gap structure

---

**Preface**

Three authors of this paper, namely Rustem Khasanov, Alexander Shengelaya, and Hugo Keller, had the opportunity to work with Alex Müller during his stay at the University of Zürich. By being specialized in measurements of the magnetic penetration depth ($\lambda$) by means of muon-spin rotation/relaxation and magnetization techniques, we try to test the prediction of Alex Müller that the symmetry of the superconducting order parameter in cuprate high-temperature superconductors (HTSs) changes "from purely $d$ at the surface to more $s$ inside" (Müller, K.A. *Phil. Mag. Lett.* **2002**, *82*, 279–288). As a result of our studies, complex order parameters were detected in $\lambda(T)$ measurements for hole-doped HTSs such as La$_{1.83}$Sr$_{0.17}$CuO$_4$, YBa$_2$Cu$_3$O$_{7-\delta}$, and YBa$_2$Cu$_4$O$_8$. Here, we present evidence that the mixed order parameter symmetry is realized in Sr$_{0.9}$La$_{0.1}$CuO$_2$, i.e., in a superconductor belonging to the family of electron-doped cuprate HTSs.

## 1. Introduction

The order parameter in cuprate high-temperature superconductors (HTSs) is generally considered to be of pure *d*-wave symmetry. Pertinent evidence for this assumption for both the electron- and the hole-doped classes of HTSs stems from experiments where mainly surface phenomena are probed (e.g., angular resolved photoemission [1–4] or tricrystal experiments [5–8]). On the other hand, experimental data obtained by using techniques that probe the bulk of the material, such as nuclear magnetic resonance [9], Raman scattering [10,11], neutron crystal-field spectroscopy [12–14], and muon-spin rotation/relaxation ($\mu$SR) [15–17] provide strong evidence for the presence of a substantial *s*-wave component in the order parameter. Based on these experimental findings and on an earlier idea that in HTSs, two superconducting condensates with different order parameter symmetries (*s*- and *d*-wave) coexist [18–20], Müller [21–24] proposed a scenario in which the order parameter symmetry in HTSs changes from primarily *d*-wave at the surface to more $d + s$-wave in the bulk. At first glance, this scenario seems to contradict the accepted possible symmetries of the order parameter in HTSs [7]. However, by applying an interacting boson-model used in nuclear physics theory to the $D_{4h}$ symmetry of HTSs, Iachello [25,26] demonstrated that a crossover from a *d*-wave order parameter symmetry at the surface to a $d + s$-wave symmetry in the bulk is indeed possible from a group theoretical point of view.

This scenario [21,22] can be directly tested by investigating the temperature dependence of the magnetic penetration depth $\lambda$ near the surface and in the bulk of an HTS, since the behavior of $\lambda^{-2}(T)$ for a *d*-wave and a *s*-wave superconductor differs considerably. An isotropic *s*-wave pairing state leads to an almost constant value of the superfluid density $\rho_s \propto \lambda^{-2}$ for $T \leq 0.3 T_c$ [27–30], while the presence of nodes in the gap gives rise to a continuum of low lying excitations, resulting in a linear temperature dependence of $\lambda^{-2}(T)$ at low temperatures [30–33].

Here, we report on a magnetization study of the magnetic penetration depth near the surface and in the bulk of the electron-doped HTS $Sr_{0.9}La_{0.1}CuO_2$. The in-plane ($\lambda_{ab}$) and the out-of-plane ($\lambda_c$) components of the magnetic penetration depth were extracted from measurements of the AC susceptibility on a *c*-axis oriented powder sample in the Meissner state and analyzed within a two-gap $s + d$-wave scenario. The present results are compared with previous muon-spin rotation ($\mu$SR) experiments on $Sr_{0.9}La_{0.1}CuO_2$ [34], which probe the magnetic penetration depth in the bulk of the sample.

## 2. Experimental Details

Details on the sample preparation for $Sr_{0.9}La_{0.1}CuO_2$ can be found elsewhere [35]. The $Sr_{0.9}La_{0.1}CuO_2$ sample used in the present study was the same as measured by means of $\mu$SR in [34].

The sintered $Sr_{0.9}La_{0.1}CuO_2$ sample was grounded for about 20 min in order to obtain very small grains needed for the determination of the magnetic penetration depth from magnetic susceptibility measurements. In order to perform measurements of the in-plane and the out-of plane components of the magnetic penetration depth, samples were *c*-axis oriented in a static 9T magnetic field and cured at elevated temperatures in epoxy resin. In total, four *c*-axis oriented samples were prepared and measured.

The grain size distribution of the powder was determined by analyzing scanning electron microscope photographs (see Figure 1a). The measured particle size distribution $N(R)$ ($R$ is the grain radius) is shown in Figure 1b.

The AC magnetization ($M_{AC}$) experiments were performed by using a commercial PPMS Quantum Design magnetometer ($\mu_0 H_{AC} = 0.3$ mT, $\nu = 333$ Hz) at temperatures between 1.75 K and 50 K. In two sets of experiments, the AC field was applied parallel and perpendicular to the *c*-axis. The absence of weak links between grains was confirmed for both field orientations by the linear magnetic field dependence of $M_{AC}$ at $T = 10$ K for AC fields ranging from 0.1 to 1.0 mT and frequencies between 49 and 599 Hz.

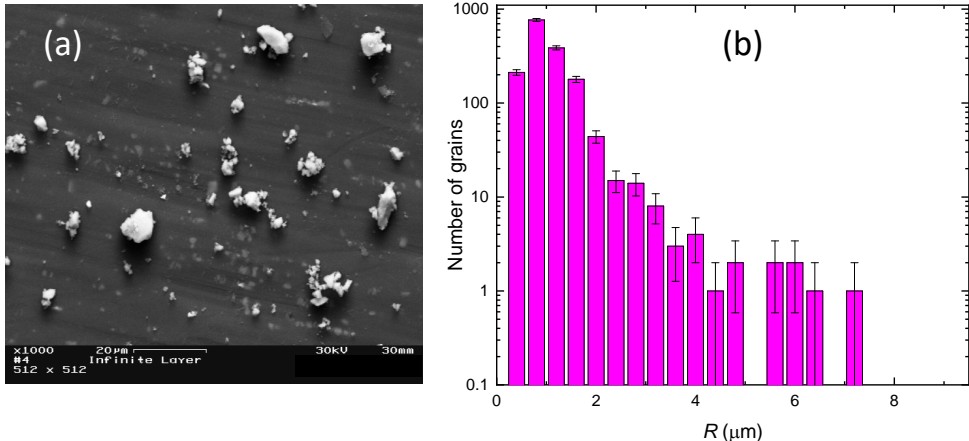

**Figure 1.** (**a**) An example of the scanning electron microscope photograph of the powdered $Sr_{0.9}La_{0.1}CuO_2$ sample. (**b**) The grain size distribution $N(R)$ determined from scanning electron microscope photographs. The thin vertical lines represent the statistical errors ($\pm\sqrt{N(R)}$).

## 3. Results and Discussion

The temperature dependence of the magnetic penetration depth was extracted from the measured $M_{AC}$ by using the Shoenberg formula [36], modified for the known grain size distribution $N(R)$ [37]:

$$\chi(T)_{\|,\perp} = -\frac{3}{2}\int_0^\infty \left(1 - \frac{3\lambda_{\|,\perp}^*(T)}{R}\coth\frac{R}{\lambda_{\|,\perp}^*(T)} + \frac{3[\lambda_{\|,\perp}^*(T)]^2}{R^2}\right)N(R)R^3\mathrm{d}R / \int_0^\infty N(R)R^3\mathrm{d}R. \quad (1)$$

Here, $\chi_{\|,\perp} = M_{AC}\cdot\rho_X/m$ ($m$ is the sample mass, and $\rho_X$ is the X-ray density of $Sr_{0.9}La_{0.1}CuO_2$) is the volume susceptibility, and $\lambda_{\|,\perp}^*$ is the effective magnetic penetration depth for the magnetic field applied parallel ($\|$) and perpendicular ($\perp$) to the $c$-axis. In order to determine $\lambda_{ab}$ and $\lambda_c$ from the measured $\lambda_{\|,\perp}^*$, we followed the procedure of Porch et al. [37]:

(i)   When a small magnetic field is applied along the $c$-axis, the screening currents flow in the $ab$-plane, decaying at a distance $\lambda_{ab}$ from the grain surface, so that $\lambda_{\|}^* = \lambda_{ab}$.

(ii)   With the magnetic field applied perpendicular to the $c$-axis, the screening currents flow within the $ab$-plane and along the $c$-axis, thus implying that both components ($\lambda_{ab}$ and $\lambda_c$) enter the measured AC magnetization. For $\lambda_c \gg \lambda_{ab}$ (which is generally the case for highly anisotropic HTSs), the effective penetration depth $\lambda_\perp^*$ is mainly determined by the out-of-plane component, and for grains of arbitrary size, the relation $\lambda_\perp^* \simeq 0.7\lambda_c$ holds [37].

Bearing this in mind, the temperature dependencies of the in-plane and the out-of plane components of the magnetic penetration depth are determined as:

$$\lambda_{ab}(T) = \lambda_{\|}^* \quad \text{and} \quad \lambda_c = 1.43\lambda_\perp^*,$$

respectively.

The resulting temperature dependencies of $\lambda_{ab}^{-2}$ and $\lambda_c^{-2}$ are shown in Figure 2.

The zero-temperature values of $\lambda_{ab}$ and $\lambda_c$ were obtained by a linear extrapolation of $\lambda_{ab,c}^{-2}(T)$ for $T < 5$ K to $T = 0$, yielding $\lambda_{ab}(0) \simeq 157$ nm and $\lambda_c(0) \simeq 1140$ nm. An uncertainty in the absolute values of $\lambda_{ab,c}$ was considered by taking into account the statistical nature of the grain size distribution ($N(R) \pm \sqrt{N(R)}$; see Figure 1b), which resulted in a relative error of about $\sim$7% for both $\lambda_{ab}$ and $\lambda_c$.

An additional source of uncertainty stemmed from the deviation of the grain shapes from the spherical one. Assuming a small deviation of the demagnetization factor ($1/3 \pm 10\%$), the relative error of $\lambda_{ab,c}(0)$ was of the order of 3%. Taking both sources of errors into account yielded: $\lambda_{ab}(0) = 157(15)$ nm and $\lambda_c(0) = 1140(100)$ nm. The value of $\lambda_{ab}(0)$ obtained here was in a good agreement with the $\lambda_{ab}(0) = 147(7)$ nm reported by Kim et al. [38] based on the analysis of reversible magnetization data.

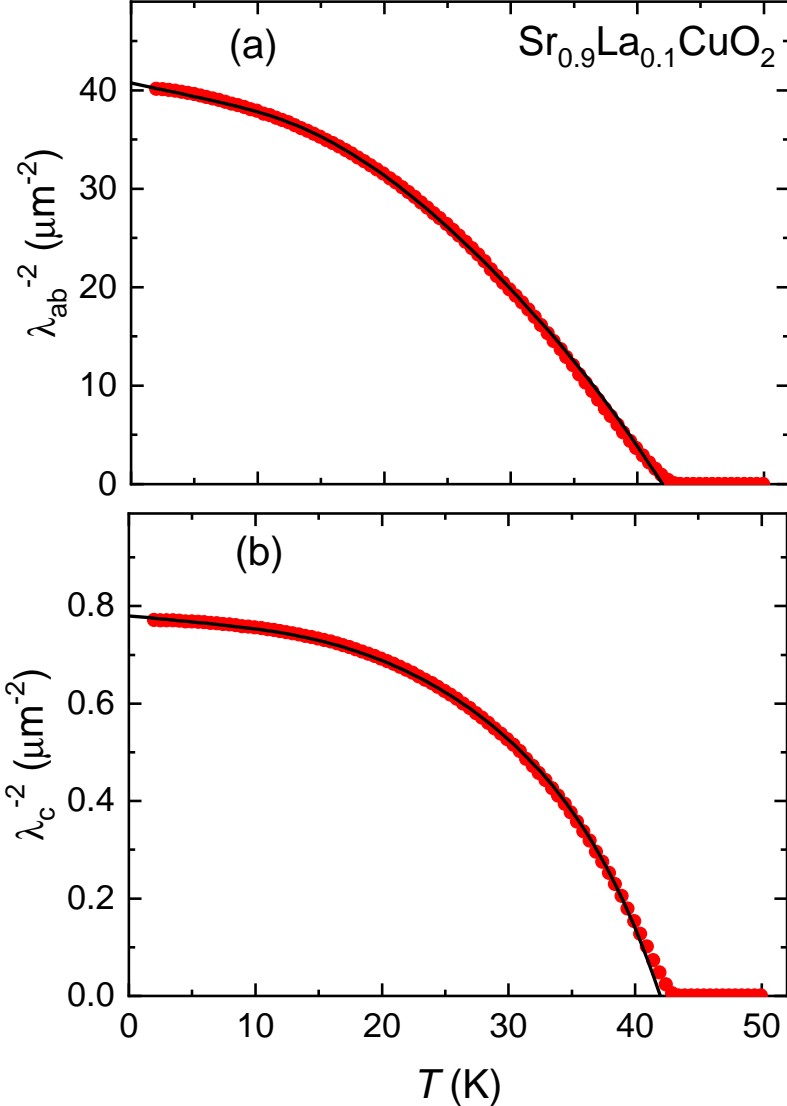

**Figure 2.** Temperature dependencies of $\lambda_{ab}^{-2}$ (**a**) and $\lambda_c^{-2}$ (**b**) for $Sr_{0.9}La_{0.1}CuO_2$ extracted from the measured $M_{AC}(T)$ by using Equation (1). Solid lines represent fits with the two-gap $s + d$-wave model. $\lambda_{ab}^{-2}(T)$ and $\lambda_c^{-2}(T)$ were analyzed simultaneously by means of Equation (2) with $\omega_{ab}$, $\omega_c$, $\lambda_{ab}(0)$, and $\lambda_c(0)$ as the individual fitting parameters and common $s$-wave and anisotropic $d$-wave gap functions as described by Equations (4)–(6). See the text for details.

In order to test the predictions of [18–22] and in analogy to our previous results on cuprate HTSs [15–17,34], the experimental data presented in Figure 2 were analyzed by decomposing $\lambda_{ab}^{-2}(T)$ and $\lambda_c^{-2}(T)$ into two contributions with *s*-wave and *d*-wave symmetry [39]:

$$\frac{\lambda_{ab,c}^{-2}(T)}{\lambda_{ab,c}^{-2}(0)} = \omega_{ab,c} \cdot \frac{\lambda_{ab,c}^{-2}(T,\Delta^s)}{\lambda_{ab,c}^{-2}(0,\Delta^s)} + (1 - \omega_{ab,c}) \cdot \frac{\lambda_{ab,c}^{-2}(T,\Delta^d)}{\lambda_{ab,c}^{-2}(0,\Delta^d)}. \tag{2}$$

Here, $\Delta^s$ and $\Delta^d$ denote the *s*-wave and the *d*-wave gap, respectively, and $\omega_{ab,c}$ is the weighting factor ($0 \leq \omega_{ab,c} \leq 1$), representing the relative contribution of the *s*-wave gap to $\lambda_{ab,c}^{-2}$. Both (*s*- and *d*-wave) components can be expressed by [16]:

$$\frac{\lambda_{ab,c}^{-2}(T,\Delta^{s,d})}{\lambda_{ab,c}^{-2}(0,\Delta^{s,d})} = 1 + \frac{1}{\pi} \int_0^{2\pi} \int_{\Delta^{s,d}(T,\varphi)}^{\infty} \left(\frac{\partial f}{\partial E}\right) \frac{E}{\sqrt{E^2 - \Delta^{s,d}(T,\varphi)^2}} \, dE d\varphi . \tag{3}$$

Here, $f = [1 + \exp(E/k_B T)]^{-1}$ is the Fermi function, $\varphi$ is the angle along the Fermi surface ($\varphi = \pi/4$ corresponds to a zone diagonal), and:

$$\Delta^{s,d}(T,\varphi) = \Delta_0^{s,d} \, \delta(T/T_c) \, g^{s,d}(\varphi). \tag{4}$$

Here, $\Delta_0^{s,d}$ is the maximum value of the gap at $T = 0$. The temperature dependence of the gap is approximated by $\delta(T/T_c) = \tanh\{1.82[1.018(T_c/T - 1)]^{0.51}\}$ [40,41]. The function $g^{s,d}(\varphi)$ describes the angular dependence of the gap and is given by:

$$g^s(\varphi) = 1 \tag{5}$$

for the *s*-wave gap and:

$$g^{d_{An}}(\varphi) = \frac{3\sqrt{3}a}{2} \frac{\cos 2\varphi}{(1 + a\cos^2 2\varphi)^{3/2}} \tag{6}$$

for the anisotropic *d*-wave gap [42] (*a* is a constant). We want to stress that series of experimental [3,43] and theoretical works [42] suggest that the angular dependence of the gap in the electron-doped HTSs differs significantly from the simple functional form $\Delta_0 \cos 2\varphi$, observed for various hole-doped HTSs, and has the so-called anisotropic *d*-wave symmetry (with the gap maximum in between the nodal and the antinodal points on the Fermi surface).

The measured $\lambda_{ab}^{-2}(T)$ and $\lambda_c^{-2}(T)$ displayed in Figure 2 were analyzed simultaneously by means of Equation (2) with $\omega_{ab}$, $\omega_c$, $\lambda_{ab}(0)$, and $\lambda_c(0)$ as the individual fitting parameters and common *s*-wave and anisotropic *d*-wave gap functions, as described by Equations (4)–(6). The results of the analysis are summarized in Figure 2 and Table 1.

Panels (a) and (b) of Figure 3 represent the angular dependencies of the *s* and the $d_{An}$ superconducting energy gaps at $T = 0$. The solid blue and the red lines in Figure 3c correspond to the individual *s*-wave and *d*-wave contributions, respectively.

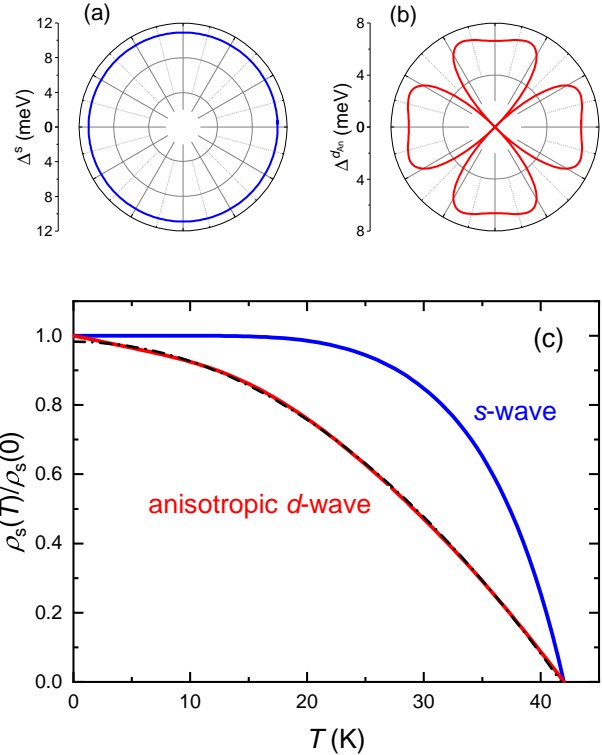

**Figure 3.** (**a**) The angular dependence of the *s*-wave gap at $T = 0$ ($\Delta_0^s \cdot g^s(\varphi)$; see Equation (5) and Table 1). (**b**) The angular dependence of the anisotropic *d*-wave gap ($\Delta_0^d \cdot g^d(\varphi)$; Equation (6) and Table 1). (**c**) Contributions of the *s*-wave (blue line) and the anisotropic *d*-wave (red line) gaps to the superfluid density ($\rho_s \propto \lambda^{-2}$) obtained by means of Equation (3). The dashed-dotted line represents the $T^2$ behavior, which is generally observed in various electron-doped HTSs (see [30] and the references therein).

From the results presented in Figures 2 and 3 and Table 1, the following important points emerge:

(i) The absolute value of the *s*-wave gap is larger than the maximum value of the anisotropic *d*-wave gap (Table 1 and Panels (a) and (b) of Figure 3) with $2\Delta_0^s/k_BT_c = 6.02(6)$ and $2\Delta_0^{d_{An}}/k_BT_c = 3.89(3)$, respectively ($T_c \simeq 42$ K). This implies that in electron-doped $Sr_{0.9}La_{0.1}CuO_2$, the *s*-wave component of the order parameter is the dominant one. This agrees with the results of small-angle neutron scattering experiments revealing that at fields higher than 1.5 T, the superfluid density of $Sr_{0.9}La_{0.1}CuO_2$ is determined entirely by the *s*-wave component of the order parameter [44].

(ii) Point (i) is in contrast to hole-doped cuprate HTSs where $\Delta_0^s$ is always smaller than $\Delta_0^d$ [10,15–17,45].

(iii) The temperature dependence of the anisotropic *d*-wave contribution to the superfluid density (solid red line in Figure 3c) is very close to the quadratic ($T^2$) dependence (dash-dotted line in Figure 3c), which is often observed in various electron-doped HTSs (see [30] and the references therein). Generally, the $T^2$ behavior is attributed to a "dirty" *d*-wave scenario and is explained by impurity scattering of the carriers. However, it is difficult to explain how an order parameter that changes sign persists in the dirty limit, since any scattering centers would act as pair breakers [46]. Therefore, we believe that the anisotropic *d*-wave approach is more appropriate for electron-doped HTSs.

(iv) For $\lambda_{ab}^{-2}(T)$, the *s*-wave contribution to the superfluid density is almost negligible ($\omega_{ab} = 0.04$), whereas for $\lambda_c^{-2}(T)$, it is substantial ($\omega_c = 0.54$) (see Table 1). Bearing in mind that our experiments were performed in the Meissner state, the different behavior of $\lambda_{ab}^{-2}(T)$ and $\lambda_c^{-2}(T)$ can be explained within the scenario proposed by Müller [21,22]. Since $\lambda_{ab}(0)$ is rather small (see Table 1), one can

assume that its temperature dependence is mainly determined by surface properties and therefore follows the one expected for a *d*-wave superconductor. In contrast, $\lambda_c(0)$ is almost a factor 10 larger than $\lambda_{ab}(0)$, and thus, $\lambda_c^{-2}(T)$ contains contributions from both the surface and the bulk (mixed $s + d$-wave order parameter).

**Table 1.** Summary of the analysis of $\lambda_{ab}^{-2}(T)$ and $\lambda_c^{-2}(T)$ for $Sr_{0.9}La_{0.1}CuO_2$ by means of Equation (2). The absolute errors of $\lambda_{ab,c}(0)$ account for the uncertainties in the grain size distribution $N(R) \pm \sqrt{N(R)}$ and that of the demagnetization factor $1/3 \pm 10\%$; see the text for details. TF-$\mu$SR, denotes the transverse-field muon-spin rotation/relaxation (TF-$\mu$SR) experiments.

| Method | Quantity | $\Delta_0^s$ (meV) | $\Delta_0^{d_{An}}$ (meV) | $a$ | $\omega_{ab,c}$ | $\lambda_{ab,c}(0)$ (nm) |
|---|---|---|---|---|---|---|
| ACsusc. | $\lambda_{ab}^{-2}(T)$ $\lambda_c^{-2}(T)$ | 10.9(1) | 7.03(6) | 0.90(2) | 0.04(2) 0.54(2) | 157(15) 1140(100) |
| TF-$\mu$SR [34] | $\lambda_{ab}^{-2}(T)$ | 10.9 [a] | 7.03 [a] | 0.90 [a] | 0.72(4) | 93(2) |

[a] From the AC susceptibility data.

Additional arguments pointing to the validity of this scenario [21,22] come from the comparison of the in-plane magnetic penetration depth $\lambda_{ab}(T)$ measured near the surface in this work with that determined in $\mu$SR experiments on a similar $Sr_{0.9}La_{0.1}CuO_2$ sample [34] (see Figure 4). Note that $\mu$SR is a powerful technique to probe the magnetic penetration depth in the bulk of a superconductor in the vortex state [31,47]. It is evident that $\lambda_{ab}^{-2}(T)$ measured near the surface decreases more strongly with increasing temperature than that obtained in the bulk. On the other hand, $\lambda_{ab}^{-2}(T)/\lambda_{ab}^{-2}(0)$ ($\mu$SR) is relatively close to $\lambda_c^{-2}(T)/\lambda_c^{-2}(0)$ determined by AC magnetization experiments, thus indicating that $\lambda_c^{-2}(T)$ is mainly governed by bulk properties. Another interesting issue stems from the comparison of the absolute $\lambda_{ab}$ values obtained in the bulk and the surface sensitive experiments. The analysis of $\lambda_{ab}^{-2}(T)$ ($\mu$SR) within the above described scheme, with $\Delta_0^s$ and $\Delta_0^{d_{An}}$ fixed to the values determined from the AC susceptibility experiments, results in $\lambda_{ab}(0) = 93(2)$ nm (see Table 1), which is more than 50% shorter than $\lambda_{ab}(0) = 157(15)$ nm obtained near the surface. We may assume, therefore, that the difference in the absolute values and the temperature dependencies of the bulk and the surface $\lambda_{ab}$ can be explained within the scenario proposed in [21,22] and is caused by a substantial reduction of the *s*-wave contribution near the surface ($\omega_{ab} \simeq 4\%$) in comparison with that in the bulk ($\omega_{ab} \simeq 72\%$; see Table 1).

To the best of our knowledge, the above presented results give a first example of different order parameter symmetries near the surface and in the bulk in electron-doped high-temperature cuprate superconductors. As for the hole-doped representatives of HTSs, the comprehensive analysis was made in a series of works of K.A. Müller [18–24]. By comparing the results of the "surface" and the "bulk" sensitive experiments, it was concluded that the superconducting order parameter in the hole-doped cuprate HTSs changes from purely *d* at the surface to the mixture of *s* and *d* in the bulk. The theory explanation was given by Iachello [25,26], based on purely symmetry considerations and in analogy with atomic nuclei. The model consists of *s* and *d* pairs (approximated as bosons) in a two-dimensional Fermi system with a surface. The transition takes place between the phase where only one type of boson condensate to the phase consisting of a mixture of two type of bosons.

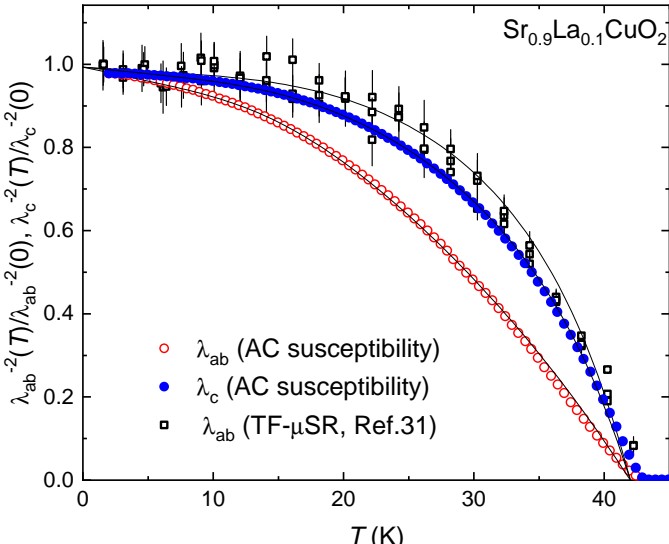

**Figure 4.** The normalized superfluid density $\lambda_{ab}^{-2}(T)/\lambda_{ab}^{-2}(0)$ (open circles and squares) and $\lambda_c^{-2}(T)/\lambda_c^{-2}(0)$ (closed circles) obtained in the present study (closed and open circles) and by the transverse-field $\mu$SR experiments (open squares) in [34]. The solid lines correspond to fits by means of Equation (2) with the parameters summarized in Table 1.

## 4. Conclusions

In conclusion, the temperature dependence of the in-plane ($\lambda_{ab}$) and the out-of-plane ($\lambda_c$) components of the magnetic penetration depth of $Sr_{0.9}La_{0.1}CuO_2$ were determined in the Meissner state from AC susceptibility measurements. The temperature dependence of $\lambda_{ab}^{-2}$ is well described by assuming that the superconducting order parameter is mainly of $d$-wave symmetry ($\simeq 96\%$) with $\lambda_{ab}(0) = 157(15)$ nm. The out-of-plane component was found to be much longer, $\lambda_c(0) \simeq 1140(100)$ nm. The temperature dependence of $\lambda_c^{-2}$ is in accordance with a mixed $s + d$-wave order parameter with a substantial $s$-wave component of more than 50%. A comparison of $\lambda_{ab}^{-2}(T)$ reported in this work with that obtained in the bulk by $\mu$SR [34] reveals that the $s$-wave component of the order parameter is strongly suppressed near the surface of the superconductor, associated with a substantial reduction of the superfluid density by more than a factor of two. The results presented here are consistent with the scenario of a complex mixed $s + d$-wave symmetry order parameter proposed by Müller [18–24]. In particular, the prediction of a strongly suppressed $s$-wave component near the surface was confirmed experimentally. This study clearly demonstrates that special care must be taken when experimental results obtained by surface sensitive and bulk sensitive techniques are compared, since they do not necessarily probe the same properties of high-temperature superconductors.

**Author Contributions:** R.K., A.S., and H.K. specified the topic of the studies. R.K. performed the experiment, analyzed the data, and wrote the manuscript. R.B. performed the electron-microscopy experiments. A.S. and H.K. contributed to finalizing the manuscript. All authors have read and agreed to the published version of the manuscript.

**Funding:** This work was supported by the Swiss National Science Foundation, by the K. Alex Müller Foundation, and in part by SCOPES Grant No. IB7420-110784.

**Acknowledgments:** The authors are grateful to A. Bussmann-Holder and K.A. Müller for stimulating discussions. D.J. Jang and S.-I. Lee are acknowledged for providing the samples used for the experiments.

**Conflicts of Interest:** The authors declare no conflict of interest.

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
