# Peer review of "Suppression of the s-Wave Order Parameter Near the Surface of the Infinite-Layer Electron-Doped Cuprate Superconductor Sr0.9La0.1CuO2"

_condensedmatter, doi:10.3390/condmat5030050_

Round 1

Reviewer 1 Report

The Authors provide experimental support to the scenario that in high temperature cuprate superconductors the symmetry of the order parameter changes from  (nearly) purely d-wave at the surface of the material to mixed s+d (fifty-fifty) in the bulk of the system.

I think the paper presents an interesting piece of work. It is also well written. I like the conclusion that surface or bulk sensitive probes do not necessarily measure the same properties of the system.

The argument that their magnetically measured ab-penetration depth being shorter then the c-axis thus contains mainly a surface contribution is not as appealing as the authors suggest. I understand the argument with simulation, but is the procedure unique? Do the Authors have some additional comments on that? What are the possible physical mechanisms behind such depth-dependent symmetry changes? The short discussion of these is, I think necessary before the paper is published.

I have also another remark related to the figure 4. Neither the legend nor the caption to this figure is acceptable. In the caption the Authors have used twice the term "closed circles" to denote two different components of the penetration depth. The symbols in the legend are a way too small and again identical, even though they should describe different data.

Author Response

The answer to the comments and criticisms of the first Referee are given in the file "Answer_to_the_first_Referee_160720.pdf" attached to the resubmission.

Reviewer 2 Report

This is a brief study of the infinite layer electron-doped cuprate superconductor Sr0.9La0.1CuO2. It has a clear aim in testing the hypothesis that the superconducting gap changes in proximity to the surface, uses appropriate experimental methods and robustly analyses the data. The contents are very clearly presented and I recommend publication without significant change.

There are typos in Refs 9 and 15 with incorrect subscripts.

Author Response

The answer to the comments and criticisms of the second Referee are given in the file "Answer_to_the_second_Referee_160720.pdf" attached to the resubmission.

Reviewer 3 Report

Referee's report on condensedmatter-866645

==============================

The manuscript reports a substantial difference between the order parameter of high-Tc superconductors at the surface and in the bulk. In principle I could recommend the manuscript for publications, but there are several minor points which could be improved:

(1) Especially on this occasion it would be nice to include the paper by KA Mueller mentioned in the preface, line 6, in the list of references, as a citation with a title.

(2) For high-Tc superconductors they use HTS's. I believe that HTSs would be better and is preferred.

(3) Eq. (1) is the Shoenberg formula which is little known, so the citation is missing in line 2 above Eq. (1).

(4) A better explanation of thin arrows in part (b) of Fig. 1 is needed.

(5) Data points and lines in Fig. 2 should be better explained to the reader.

(6) In "Acknowledgments" the closing paranthesis at the end is obsolete.

It is my pleasure to recommend minor revision to implement these changes.

Author Response

The answer to the comments and criticisms of the third Referee are given in the file "Answer_to_the_third_Referee_160720.pdf" attached to the resubmission.

Reviewer 4 Report

The paper deals with experimental measurements of the penetration depth and a theoretical analysis to interpret the results leading to the s-component of the superconducting gap is present in the bulk but vanishes near the surface. I like papers with this combination of theory and experiments and believe that there are enough results to be published but there are a few issues that deserve attention to the authors.

1- Fig.1a shows a typical picture of the powder with some grains and Fig.1b shows the distribution of these particles N(R). I could not see even a single grain in Fig.1a) that could fit in the distribution. I suggest the authors to show at least an example of how one single grain is measured. This is important because Eq.1 relies on N(R) and the derived values of both λ .

2- There is a confusing sentence in the end of the paragraph after Eq. 1: "For λ_c >> λ_ab (which is generally the case for highly anisotropic HTS’s) the effective penetration depth λ_⊥ is mainly determined by the out-of-plane component and for grains of arbitrary size the relation λ_⊥ = 0.7λ_c holds[33]". The reader does not know which case, λ_c >> λ_ab or λ_⊥ = 0.7λ_c is used in the calculations and plotted in Fig.2.

3- The temperature dependence (in line 94) is based on ref. 36 and 37 that deal with BCS type gaps what is not the case of cuprates. Therefore I think that this point needs an explanation.

4- In lines 102-102 we read that W_ab,c and λ_ab,c are fitting parameters fitting parameters but the gaps Δ^0_ab,c are not mentioned. I suppose they are derived from Eq. 3 but this should be more clearly explained.

5- The conclusion that try to support Ref,18-22 seems to be heavily dependent on the fitting parameters W_ab,c, specially W_ab =0.04 given in table1. Consequently the readers will wonder if we could obtain others fitting parameters, specially with a larger W_ab and perhaps different λ_ab,c , that could also reproduce Fig.2. A larger W_ab would imply in a different conclusion. Another minor points are: Fig. 3 has some errors in the legend. Ref.33 has a misspelling In summary I think that the authors should pay attention to the above points to improve their manuscript.

Author Response

The answer to the comments and criticisms of the fourth Referee are given in the file "Answer_to_the_fourth_Referee_160720.pdf" attached to the resubmission.

Round 2

Reviewer 1 Report

The Authors have adequately answered the questions and the corrected caption to Fig. 4 is good enough to understand the message of it.

I thus propose the publication of the paper in the present form.

Reviewer 3 Report

The authors improved their presentation and addressed issues mentioned by the referees 1 and 4. There are no further suggestions.

It is my pleasure to recommend publication without changes.

Reviewer 4 Report

I have looked at this new version and am satisfied with the modifications.
This version of the paper may be accepted and published.
Yours,